# Abiotic Stress Triggers the Expression of Genes Involved in Protein Storage Vacuole and Exocyst-Mediated Routes

**DOI:** 10.3390/ijms221910644

**Published:** 2021-09-30

**Authors:** João Neves, Miguel Sampaio, Ana Séneca, Susana Pereira, José Pissarra, Cláudia Pereira

**Affiliations:** 1Faculdade de Ciências, Universidade do Porto, Rua do Campo Alegre, s/n°, 4169-007 Porto, Portugal; jneves@fc.up.pt (J.N.); miguelfsampaio@hotmail.com (M.S.); aseneca@fc.up.pt (A.S.); mspereir@fc.up.pt (S.P.); jpissarr@fc.up.pt (J.P.); 2GreenUPorto-Sustainable Agrifood Production Research Centre, Department of Biology, Faculty of Sciences, University of Porto, Rua do Campo Alegre, s/n°, 4169-007 Porto, Portugal

**Keywords:** abiotic stress, endomembranes, vacuolar trafficking, exocyst, SNAREs, gene expression, ultrastructure

## Abstract

Adverse conditions caused by abiotic stress modulate plant development and growth by altering morphological and cellular mechanisms. Plants’ responses/adaptations to stress often involve changes in the distribution and sorting of specific proteins and molecules. Still, little attention has been given to the molecular mechanisms controlling these rearrangements. We tested the hypothesis that plants respond to stress by remodelling their endomembranes and adapting their trafficking pathways. We focused on the molecular machinery behind organelle biogenesis and protein trafficking under abiotic stress conditions, evaluating their effects at the subcellular level, by looking at ultrastructural changes and measuring the expression levels of genes involved in well-known intracellular routes. The results point to a differential response of the endomembrane system, showing that the genes involved in the pathway to the Protein Storage Vacuole and the exocyst-mediated routes are upregulated. In contrast, the ones involved in the route to the Lytic Vacuole are downregulated. These changes are accompanied by morphological alterations of endomembrane compartments. The data obtained demonstrate that plants’ response to abiotic stress involves the differential expression of genes related to protein trafficking machinery, which can be connected to the activation/deactivation of specific intracellular sorting pathways and lead to alterations in the cell ultrastructure.

## 1. Introduction

Nowadays, climate change stands as a major threat to human well-being and survival, as it is the leading cause of crop collapses worldwide, leading to food insecurity and scarcity [1]. As sessile organisms, plants evolved to adapt to and take advantage of changes in climate and environment changes [2]. However, abiotic stresses, particularly drought, soil salinity and extreme temperature, are often interrelated and the main inducers of oxidative stress in cells, causing cellular damage [3]. These diverse environmental stresses often activate signals and pathways involved in similar cellular responses: overexpression of antioxidants, accumulation of solutes, changes in protein trafficking and endomembrane remodelling [4,5,6]. In recent years, high throughput screening techniques, such as microarrays and RNA sequencing, identified many stress-related genes. These techniques provide us with important information and suggest genes among which it is possible to identify new markers for assisted selection of varieties resistant to stress. Changes in the transcriptome are still the result of a complex series of events and understanding the stress response mechanism is only partial. One of the most relevant mechanisms occurs at the endomembrane level, particularly in inter-organellar communications [7,8]. Identifying the specific roles of each actor in the game turns out to be an essential factor for the genetic improvement of plants because the positive adaptation probably depends on synergistic effects and balanced interactions among proteins that are commonly not related [9]. Recent experimental evidence [9,10] suggests that several classes of proteins (such as Aquaporins, SNARES, ATPase pumps or channels) can control specific membrane transport events, leading to important events of cell reorganisation in adverse environmental conditions. Many studies have supported the idea that endomembrane trafficking is tightly linked to stress signalling pathways, unfortunately without a deeper understanding of the underlying mechanisms [11]. The research effort in the plant stress field is formidable and the literature is rich in data. Several research groups found exciting connections between stress tolerance and membrane rearrangements, as was the case of a potassium channel selectively accumulated on small vacuoles not observed before [12] and sufficient to confer stress tolerance when overexpressed. Nonetheless, the connection between the membrane architecture and stress tolerance has not been sufficiently investigated. In the last decade, the outburst of data on molecular mechanisms involved in protein trafficking has outlined subtle balances between the transport routes [13,14]. The physiology of plants and crops subjected to stress has been an issue of several studies through recent decades. Still, no or little attention has been given to the molecular mechanisms controlling these rearrangements. Transport and compartmentalisation events of a large amount of material without the involvement of the classic vesicle traffic have escaped full characterisation. One example is the recently characterised vacuolar route mediated by the PSI (Plant Specific Insert), which follows an alternative pathway independent of the Golgi [15]. This process is generically described as unconventional trafficking, and several examples have been reported without a deep mechanism investigation [16,17]. Many of these unconventional routes are associated with development and stress-induced conditions since cells need to rapidly adapt their trafficking machinery to a new, challenging scenario. Moreover, transport of cargo molecules between compartments is thought to occur mainly by vesicle shuttles (i.e., transport vesicles). The cytoskeleton has a role in facilitating vesicle transport from one compartment to another [18]. The relevance of this “shuttle transport” might attain more relevant outlines in the context of cellular reorganisation when the plant is facing adverse conditions.

Identifying the specific roles of each player in this game turns out to be an important factor for plant genetic improvement since positive adaptation probably depends on synergistic effects and balanced interactions usually not considered to be related [19]. In this context, this work aims to understand the mechanisms of reorganisation and remodelling of plant cell endomembranes in response to abiotic stress. The main objective was to identify and cluster genes involved in several intracellular pathways according to changes in their expression in plants grown under different stress conditions. In parallel, we also aim to identify ultrastructural changes in the endomembrane system induced by stress. Our results show that genes that regulate the Protein Storage vacuole pathway are upregulated when plants grow in adverse conditions, along with genes involved in the exocytic route, indicating a positive regulation of some trafficking routes.

## 2. Results

### 2.1. Arabidopsis thaliana Germination and Growth Are Affected by Stress

*Arabidopsis thaliana* seeds were germinated in plates containing modified MS medium to simulate different abiotic stress conditions: Saline (S1 and S2), Hydric (H1 and H2), Oxidative (Ox) and metal poisoning (Zn) (Figure 1A). After 10 to 12 days of germination, the seedlings were harvested for RNA extraction, and fixation for Electron Microscopy and the macroscopic morphology was recorded (Figure 1B). In the saline and hydric stress conditions (Figure 1C-S1, S2, H1 and H2), the morphological alterations were very similar and consisted mainly of a development delay compared to control and some chlorosis in the foliar tissue. However, the salt concentrations used have a more pronounced effect on the plant growth than the water deficit conditions, resulting in a significant delay in development, which is more prominent in the higher salt concentration (S2). In the oxidative stress (Figure 1C—Ox), the plants developed normally, and in the stress induced by metal (Figure 1C—Zn), leaf chlorosis was very notorious. The length of the primary roots was also evaluated, and the only conditions where a significative decrease relative to control was observed were in H2, S1 and S2 (Figure 1D). Interestingly, the difference in root length between the two salt concentrations used is also statistically significant. This correlates with the developmental defects observed for S2 seedlings.

### 2.2. Abiotic Stress Induces Alterations at the Ultrastructural Level

Leaf (Figure 2 and Figure 3) and root (Figure 3) segments from seedlings germinated under stress conditions (control, S1, S2, H1, H2, Ox and Zn) were fixed and observed under an Electron Microscope to depict any ultrastructural changes, particularly in what concerns endomembrane organization (Figure 2A and Figure 3A).

#### 2.2.1. Ultrastructural Characterisation in Leaf Tissues

In control sections of the leaf (Figure 2B(a)), the cell structure is quite evident. Leaf cells have large central vacuoles that occupy the most cell space, and the cytoplasm and organelles are pushed to the periphery. In a closer look, the chloroplasts’ grana organisation, the distinction between stroma and thylakoid membranes, and the oval and discoid forms typical of this organelle are noticeable (Figure 2B(a)). The cell wall and the cytoplasmic membrane are well preserved and organelles, such as mitochondria, are dispersed within the cytoplasm. Additionally, it is possible to observe small plastoglobuli inclusions inside the chloroplasts.

At a low salt concentration—S1—no significant differences are observed relative to control, except for a higher accumulation of starch granules and plastoglobuli inclusions in the chloroplasts (not shown). In the S2 condition, a higher salt concentration, some plasmolysis can be observed, with the cytoplasmic membrane detached from the cell wall in some cases (Figure 2B(b)). The cytoplasm is denser when compared to control conditions, and the chloroplasts have several plastoglobuli inclusions of large dimensions. Furthermore, dilation of the thylakoid membrane can be observed (Figure 2B(b), arrows), and in several cases, some membranous structures can be observed inside chloroplasts (Figure 2B(b), arrowheads).

For the lower concentration of mannitol—H1—no changes are observed (data not shown). However, when higher concentrations of this agent are used—H2 condition—significant differences can be observed, especially in the chloroplasts (Figure 2B(c)), where a variable number of large starch grains can be observed. Still, few or no plastoglobuli are detected, contrary to the S2 condition. No plasmolysis is observed.

It is also clear that oxidative stress (Ox) affects the chloroplasts by the destruction of the lamellar system (Figure 2B(d)), with marked alterations in the stroma and thylakoid organisation, which in some cases are no longer defined (Figure 2B(d), arrows). Oxidative stress is also the condition tested where more differences in endomembrane alterations are visible (Figure 3B, arrowheads). Several membranous structures are observed in the cytoplasm and in close association with the cytoplasmic membrane (Figure 3B(c)). The Golgi Apparatus and the endoplasmic reticulum show some degree of hypertrophy with enlarged cisternae (Figure 3A(c,d), arrows).

Finally, in the stress condition induced by metal—Zn—the chloroplasts are severely affected, with marked alterations in the stroma and thylakoids membranes, which become disorganised and curved (Figure 2B(e)). Several plastoglobuli inclusions and large starch granules are also observed inside the chloroplasts.

#### 2.2.2. Ultrastructural Characterisation of Root Cells

In root cells, the large central vacuoles are not present, so the cytoplasm and organelles are not pushed against the cell wall. In the control condition, it is possible to observe a denser cytoplasm when compared to the leaves (Figure 2B(a)).

The changes regarding the endomembrane system are noticeable in the H2 and Ox conditions, especially in the endoplasmic reticulum, Golgi and associated vesicles. These alterations are evident in the H2 condition, where hypertrophy of the Golgi is observed (Figure 3B(b), arrow). Particularly evident is the swelling at the edge of the cisternae in the oxidative stress samples (Figure 3B(c,d), arrows). In these samples, many vesicles in the cell are also detected, frequently associated with the Golgi and endoplasmic reticulum (Figure 3B(d)).

### 2.3. Endomembrane Trafficking-Related Genes Are Differentially Expressed under Stress

Several endomembrane system-associated genes were selected for this study based on their translation products’ roles, localisation and involvement in intracellular pathways, namely vacuolar sorting receptors, SNAREs involved in membrane docking and fusion and exocyst and autophagy subunits (Table 1).

From the literature available, it is known that several of these gene products interact and form stable complexes. We plotted these genes’ products in a protein–protein interaction network functional enrichment analysis software—String [35]—and the interaction network between them is clear, particularly among the SNAREs used in the study (Figure 4A). EXO70 and ATG8 are far from the SNARE cluster but interact with SYP61 and VTI12, respectively. The selected VSRs also cluster together and show interactions with different members of the SNARE family. Some of these gene products have been already associated with adverse environmental conditions, but the majority were not. Thus, their expression was evaluated by qRT-PCR in plants germinated under abiotic stress and compared to control conditions. The relative expression of the different genes in the control condition is variable (Figure 4B). *AtVSR2*, *AtVTI11*, *AtSYP23* and *AtVAMP723* are clearly more expressed than the other genes and *AtEXO70* and *AtVTI12’s* relative quantity shows the lowest values. We obtained a heat map of the Pearson correlation between the normalized expression of the different genes (Figure 4C, rows) and the stresses applied (Figure 4C, columns) and simultaneously clustered this data by average linkage. The results allowed us to distinguish differences in gene expression according to the type of stresses applied, grouping them into three different stress groups, according to the z score values: (1) genes whose expression is generally positively correlated with H2, Ox and Zn and have lower expression in the control; (2) genes whose expression is generally positively correlated with S1 and H1 and also have lower expression in the control; (3) genes whose expression mostly shows no correlation with the stresses studied, but most have an increased expression in the control (z score 0.5–2) (Figure 4C, 1, 2 and 3). *AtSYP52*, *AtRMR1* and *AtEXO70* cluster in the first group, all positively correlated (z score 0.5–2), except *AtEXO70* in the Zn condition. The second group includes most of the other genes, and their expression is essentially positively regulated in H1 and S1 stress conditions, very different from the control condition. Notably, most of them show a decrease in expression in H2 stress (z score < 0). The genes clustered in the third group—*AtVSR2*, *AtSYP23*, *AtVAMP723* and *AtVTI11*—show a slight decrease or increase in their expression levels across the stresses studied but show an expression pattern that is mostly different from the control. AtVTI11’s expression is decreased relative to the control (z score < −0.5) with H2 and Zn but is similar to the control in S1 (z score > 0.5), and AtSYP23’s expression is decreased relative to the control (z score < −0.5) with H1, which is only shared with *AtSYP52*. From the data presented, different stress conditions have different effects on the expression of these genes. The results obtained were then grouped according to the gene product’s known role in different routes of the secretory pathway and putative interactor partners for better analysis (Figure 4B, coloured boxes).

#### 2.3.1. Genes Involved in the Vacuolar Pathway

Initially, a group of genes known to be involved both in the route to the Protein Storage Vacuole—*AtRMR1, AtSYP51, AtVPS45* and *AtVTI12*—and to the Lytic Vacuole—*AtVSR2*, *AtVTI11* and *AtSYP52*—was analysed. In control conditions, and considering their relative quantity in the samples, it is interesting to see that the genes involved in the route to the LV are more expressed (2–5 fold) than the ones associated with PSV routes (Figure 4B). Regarding the stress analysis, almost all the genes selected showed changes in their expression relative to the control: *AtRMR1* and *AtSYP51* are overexpressed in almost all stress conditions. In contrast, *AtVPS45* is only overexpressed in H1 and downregulated in the others, though significative differences are only found in the H2 sample (Figure 5A). The VTI12 protein also participates in the route to the PSV, and this gene is also overexpressed in all the conditions under study (Figure 6C). Its expression is almost 30 times higher than in the control. On the other hand, the genes associated with the LV pathway are mostly downregulated in all the conditions under study: *AtVSR2* is downregulated in all conditions and *AtVTI11* is downregulated in most of them (Figure 5B). *AtSYP52*, however, does not follow this tendency, showing a significant overexpression value in the Ox condition and a non-significative variability in the other stresses (Figure 5B). The results so far point to the hypothesis that the genes linked to the PSV route are positively regulated under the stress conditions tested. To confirm this tendency, two more genes encoding vacuolar sorting receptors known to participate in PSV sorting were selected—*AtRMR2* and *AtVSR1*. Interestingly, *AtRMR2* is only upregulated in the H1 condition, and its expression does not significatively change in the other stress conditions tested. At the same time, *AtVSR1* is overexpressed in all stress conditions, despite the values obtained for H2 and Ox not being significant (Figure 5C).

#### 2.3.2. SNAREs Involved in Plasma Membrane Vesicle Docking and Fusion

Several SNAREs involved in docking and fusion at the plasma membrane are associated with the opening of potassium channels and engaged in maintaining osmotic regulation. We analysed the expression of the genes encoding for SYP61, SYP121 and VAMP722. In control conditions, their relative expression is not significantly different (Figure 4B). However, in stress conditions, the expression of *AtSYP61* is higher when compared to *AtSYP121* and *AtVAMP722*. The expression of *AtVAMP722* shows no significant alteration, while that of *AtSYP61* and *AtSYP121* is tendentially enhanced throughout the study (Figure 6A). Intriguingly, the overexpression of *AtSYP61* is only significant in the S1 and H1 conditions. At the same time, in the case of the *AtSYP121,* this is true for S1, Ox and Zn and has no significant alterations in hydric stresses (Figure 6A). Additionally, we also tested the expression of an *AtVAMP722* protein homologue—*AtVAMP723*—that is found at the ER level and the gene encoding SYP23, an unusual SNARE that lacks the transmembrane domain. Not much is known regarding these two proteins aside from its localisation, and it is noteworthy to observe that both encoding genes are downregulated for all the conditions tested (Figure 6B). Given this result, we decided to also analyse the expression of *AtSYP22*, a homologue of *AtSYP23*, where the encoded protein has a transmembrane domain, and it is known to function between the PVC and the vacuole. In our study, contrary to *AtSYP23*, *AtSYP22* is upregulated in all the conditions, particularly in H1 and S1 (Figure 6B). However, in control conditions, *AtSYP23* expression is higher than that observed for *AtSYP22* (Figure 4B).

#### 2.3.3. Exocyst and Autophagy Pathways Genes

Autophagic processes are widespread in cells under stress, and the recently discovered exocyst-mediated pathway has also been associated with adverse intracellular conditions. It can be seen as an alternative autophagic/exocytic route. We chose the autophagic marker ATG8 and the exocyst subunit EXO70, along with the SNARE VTI12, which is known to mediate the docking of autophagic/exocytic vesicles with the vacuole. The autophagy/exocyst-related genes behave very differently in the control condition, where the expression of *AtATG8* is prevailing compared with the one observed for *AtVTI12* and *AtEXO70* (Figure 4B). However, when looking at the stress conditions, the expression of *AtATG8* is very low compared with *AtEXO70*, which is overexpressed in all conditions, with a minimum 8-fold higher in Zn, and a maximum expression in the H2 with roughly 20-fold more than in control (Figure 6C). The expression values obtained for the *AtATG8* are all significant, overexpressed in H1 and S1 and downregulated in H2, Ox and Zn conditions. *AtVTI12* expression matches the relative expression values for *AtEXO70*, being also upregulated in all stress conditions, with a maximum value 28-fold higher than the control in the S1 condition (Figure 6C).

## 3. Discussion

As previously discussed, during their life cycle, plants can be exposed to several adverse environmental conditions, including salinity, drought, extreme temperatures, and metal poisoning, commonly characterised as abiotic stresses. These conditions will affect several molecular, biochemical and physiological processes resulting in developmental delays and eventually senescence and cell death [36,37,38]. Because plants are immobile organisms that cannot respond to environmental stimuli like animals, developing adequate responses at the cellular level is crucial. It is known that the stress responses entail the expression of a large number of genes, involved in many different cellular processes and signalling pathways. Although several regulators (positive and negative) have been described so far, the molecular regulatory mechanisms at the basis of plant adaptation to stress remain essentially unknown.

### 3.1. Abiotic Stress Induces Endomembrane Rearrangements

It is of common sense that defects in the growth and development of plants start at the cellular level, and most frequently, changes are observed at the ultrastructural levels. Several studies focus on the ultrastructural aspects of cells under stress [39,40,41]. However, they are mainly focused on chloroplast phenotypes, as these are the most expressive, especially when looking at leaf cells with their large chloroplasts compared to other organelles. In the present study, alongside alterations in chloroplast morphology, we looked for alterations in endomembranes, particularly those related to the endoplasmic reticulum and Golgi, as these are central organelles in the endomembrane trafficking routes. Our results show different responses to the different types of stress applied. The major changes in the endomembrane system were observed in the oxidative stress condition, which is interesting since no significant differences were observed in the seedling development when compared to the control. The observed alterations in the endomembrane system do not necessarily mean that the plant’s normal development is affected. In fact, it may represent a mechanism for the cells to cope with the adverse environment they are facing, allowing the plant to develop normally. The high detail provided by the electron microscopy techniques here employed allowed us to detect a phenotype that would not have been picked in other assays. In oxidative stress conditions, both in leaves and root segments, it is possible to observe hypertrophy of the Golgi apparatus, particularly evident in the swollen edges of the cisternae in root cells. This is frequently associated with high vesiculation, indicating a highly active Golgi, a signal of intense protein trafficking. In leaf cells, the presence of many membrane formations in the cytoplasm or associated with the vacuoles is also seen repeatedly, indicating that a high degree of membrane remodelling is taking place in those cells. Regarding the chloroplast phenotype, it is possible to see, in some conditions, alterations in the organization of the thylakoid membranes organization, which may be the result of increased intracellular concentration of reactive species of oxygen that leads to the destruction of membranes and proteins, inducing a hypersensitive response [41]. In the case of excess salinity, the most noticeable feature is the presence of dilated thylakoids in the chloroplasts, which can indicate an alteration of the chemical composition of the stroma related to a more active functioning of ionic pumps [40], induced by the excess of salt to balance the quantity of ions. Contrastingly, the grana are usually arranged regarding the water deficit condition, but large starch grains can be observed inside the chloroplasts. The high amount of storage of substances found in these samples, particularly of carbohydrates, can be seen as a result of greater intracellular compartmentalisation [42] to manage the water deficit the cells are subjected to. Moreover, it is also possible that this observation is related to the impairment of sugar metabolism, which may indicate a decrease in cellular respiration. Finally, in plants grown under metal stress, major changes in the structure of the grana and intergrana are noticeable, which will undoubtedly affect photosynthesis. Taken together, the data obtained show clearly that the environmental conditions under which plants grow greatly affect all physiological processes inherent to survival and development, and that these changes start at the subcellular level, being chloroplast morphology, membrane reorganisation, high vesiculation and changes in Golgi morphology, a good readout of plant sensitivity and/or adaptation to adverse conditions.

### 3.2. Key Regulators of Endomembrane Trafficking Are Differentially Expressed under Stress

Endomembrane trafficking is essential for all eukaryotic cells as several proteins are constantly being produced and need to be delivered to the correct location to perform their function. The molecular mechanisms and effectors underlying such pathways have been extensively studied over the years, and nowadays, very detailed information is available, translated in several excellent reviews over the years [17,42,43,44,45]. The endomembrane system in plants needs to constantly adapt to the morphological demands of development or a changing environment. Thus, trafficking pathways and their associated machinery are tightly linked to stress signalling pathways for the de novo expression and/or re-location of stress-related proteins [11,23,46]. Here, we tested the expression of several endomembrane-related genes involved in different intracellular pathways in plants grown under various abiotic stress conditions. The 17 genes selected for this work were clustered in four groups: (1) related to the PSV/LV trafficking (*AtRMR1, AtRMR2, AtVSR1, AtVSR2, AtVPS45, AtSYP51, AtSYP52* and *AtVTI11*); (2) involved in membrane docking and fusion at the plasma membrane trafficking (*AtSYP61, AtSYP121, AtVAMP722*); (3) SNAREs of unknown function (*AtSYP23, AtSYP22* and *VAMP723*); and (4) genes involved in autophagy/exocytosis routes (*AtATG8, AtEXO70* and *AtVTI12*). Almost all the genes tested responded to the stress conditions, with alterations of their expression levels, which allowed us to distinguish positively regulated pathways under stress. Moreover, a differential response of the genes tested to a given stress condition is evident. The clustering analysis and the associated heat map show that the H2 condition forms a separate cluster, with most genes being negatively correlated in this stress. In fact, the seedlings’ development in this condition is greatly compromised. Another interesting cluster observed is the grouping of H1 and S1 stresses in the same branch, which is not unexpected, as salt and drought stresses induce a similar response in the cells and activate the same mechanisms [47]. The results obtained for Zn stress are also relevant, as this condition seems to trigger specific intracellular mechanisms and, in fact, it clusters in a separate branch from the salt and water stresses. It is known that the cells’ responses to metal poisoning involve altering membrane permeability, storage, and detoxification [48], correlating well with the results obtained for *AtRMR1*, *AtVSR1*, *AtSYP61* and *AtSYP121*, involved in the route to the PSV and plasma membrane-associated vesicle docking, respectively.

#### 3.2.1. Genes Involved in the Path to the PSV Are Positively Regulated under Stress

In the first group of genes, those related to vacuolar trafficking, which were overexpressed, *AtRMR1*, *AtSYP51* and *AtVTI12,* are associated with the route to the protein storage vacuole (PSV). RMR1 is a vacuolar sorting receptor found to localize to the Golgi, TGN and PSVs and mediates the transport of proteins carrying a C-terminal vacuolar sorting determinant (ctVSD). It has been implicated in sorting storage proteins to the PSV via precursor accumulating vesicles (PACs) [20,25]. On the other hand, VSR2, another type of vacuolar sorting receptor, is responsible for protein transport to the lytic vacuole and, through a different pathway and mechanism, recognizes a typical NPIR motif present at the N-terminus of vacuolar proteins and mediates trafficking through clathrin-coated vesicles (CCVs) [34,49]. In our study, the encoding gene for VSR2 is clearly downregulated for all the stress conditions analysed. Taken together, the results obtained for these two genes encoding vacuolar receptors suggest that the vacuolar transport in cells under abiotic stress conditions, such as the ones tested, is biased towards the PSV pathway. This hypothesis is quite exciting as it indicates that plant cells can shift their sorting mechanisms, allowing the accumulation of diverse molecules in the PSV to face adverse conditions. This premise is strengthened when we look at the other genes in this study. *AtSYP51* and *AtVTI12* encode different functional classes of SNARE proteins. Both showed to be positive regulators of proteins with a CtVSD in their sorting to PSVs, as was demonstrated for chitinase A [26,33,50,51,52]. Both *AtVTI12* and *AtSYP51* are clearly upregulated in all conditions tested, supporting this theory. Another evidence supporting this theory is the downregulation of *AtVTI11*, another gene coding for a SNARE, that mediates protein trafficking dependent on CCVs, thereby related to the transport to the LV [26]. Interestingly, its interaction partner—SYP52, also involved in the route to the LV [29]—does not behave in the same way. The translated gene is upregulated in the Ox and Zn conditions, with no changes in other conditions tested. This opposite behaviour of two interacting partners is quite interesting, as it opens the door to a more detailed investigation of their roles in cells under stress. Another gene that does not fit our theory is AtVPS45, which only shows a positive response in the H1 condition. This is intriguing as VPS45 has been described as a positive regulator of PSV trafficking [31] and could indicate other roles for VPS45 dependent on its interaction partners VTI12 and SYP61. Considering that it is only upregulated in cells facing a water deficit, this would be interesting to pursue in future studies. Finally, and to increase the strength of our hypothesis, we checked the expression of two additional genes encoding VSRs involved in the trafficking to the PSV—*AtVSR1* and *AtRMR2* [32,53]. Both are essentially upregulated, adding extra confidence to our theory. However, it is worth pointing out that the expression of *AtRMR2* in the H1 condition is particularly high (8-fold higher than control), while in the other conditions, it does not significantly change. This could indicate a specific mechanism occurring in water deficit conditions, particularly if we compare the result obtained for *AtRMR1* that is not statistically significant and in control conditions is more expressed than *AtRMR2*. As a whole, the results shown here point to a shift towards the PSV pathway in plants subjected to abiotic stress that seems to be independent of the type of stresses applied and should thus be considered a primary line of cell defence/adaptation to stress.

#### 3.2.2. SNAREs Involved in Osmotic Regulation at the Plasma Membrane Are Selectively Expressed

SNAREs involved in plasma membrane transport are often associated with osmotic adjustments and in the control of ionic channels, thus being a good readout for abiotic stress response. SYP121 localizes at the plasma membrane and has been implicated in stomatal closure and K+ channel activity following stress signals [27,54]. SYP121 is therefore important in the control of cellular volume and osmotic adjustment. Nevertheless, we only found significant changes in the S1, Ox and Zn conditions, indicating that the gene’s activity encoding this protein in cells undergoing water deficit may be not relevant. Alongside SYP121, we also checked SYP61, a TGN-localized syntaxin that forms a SNARE complex important for regulating plasma membrane aquaporins [10], and therefore participates in the modulation of water permeability at the plasma membrane. The gene’s expression profile is the opposite of the one verified for *AtSYP121,* being increased in H1 and S1. This correlates well with the roles described for SYP61, since the increase in gene expression in the salt and drought stresses were more evident than in oxidative or heavy metal stress. SYP61 is also known to form a complex with VTI12, and despite that VTI12 is also upregulated in the same conditions, the relative expression values are very different. VAMP722, in its turn, is a SNARE involved in biotic stress responses that forms a complex with SYP121 [24], so we asked whether it could also be implicated in abiotic stress signalling. The results indicate otherwise, as no significant changes are observed. Overall, it is possible to say that the SNAREs responsible for the transport and vesicle docking at the plasma membrane respond differently to abiotic stress. Still, more evidence is necessary to clarify their role.

#### 3.2.3. Non-Characterised SNAREs, SYP23 and VAMP723, Are Downregulated

Looking through the literature available, we came across two other SNAREs that caught our attention: SYP23 and VAMP723. Despite not being as well characterised as their homologues, these two proteins have uncommon localizations: most of the SNARE proteins are found at the post-Golgi level. However, VAMP723 is found at the endoplasmic reticulum [55], and SYP23 is a cytoplasmic protein, which lacks a transmembrane domain [56]. In our study, these two genes were downregulated in all stress conditions. The biological relevance of these two genes under abiotic stress conditions is not clear from this study, and it is worth comparing it with other members of the same family to try to understand their role in this process. SYP22 is a member of the Qa-SNAREs family, mostly found at the vacuolar membrane [57]. In contrast to SYP23, SYP22 possesses a transmembrane domain that allows its presence in the vacuole membrane. In our study, SYP22 encoding gene (AtSYP22) expression only shows a statistical significance in the S1 sample, although we might consider a tendency for a positive regulation.

#### 3.2.4. Exocytosis and/or Autophagy Mediated by the Exocyst Is Stimulated by Stress

The autophagic pathway is usually associated with alterations of the cell endomembranes in stress conditions, and recently a novel autophagic pathway, mediated by the exocyst, has also been implicated in such responses [2,55]. In this work, we also wanted to view how these two pathways behave under stress conditions and evaluate the expression of the genes coding for ATG8, EXO70 and VTI12 that mediated vesicle fusion of both pathways with the vacuole. To our surprise, *AtVTI12* showed a much higher increase in relative expression when compared to the other genes tested, in particular during salt stress (approx. 30-fold higher). This observation may be related to the diverse roles of VTI12 in mediating protein trafficking to the PSV and participation in the plant autophagic route [26,33]. This process plays a crucial role in plant adaptation to adverse conditions either derived from environmental stress or pathogen attack [33]. Another upregulated gene encodes autophagy-related protein 8 (ATG8) since it is a well-known marker for cellular autophagy [21]. The results obtained through qPCR showed significant alterations in the expression pattern of *At ATG8* under all stress conditions, but on a smaller scale when compared to *AtVTI12* or *AtEXO70*. In fact, stress seems to positively affect *AtATG8*, especially in S1 and H1. We also saw a down-regulation of *AtATG8* in H2, Ox and Zn stress conditions, which can be due to cells using other genes to respond to this kind of stress. Finally, we checked the expression of *AtEXO70*, which encodes a key player in exocyst-positive organelle (EXPO) formation. This protein has been involved in autophagy processes and the vacuolar pathway in animals and plants [56]. In our study *AtEXO70* is upregulated in all conditions at levels much higher than the ones observed for *AtATG8* or other genes in this study, except for *AtVTI12*. This should be a clear indication of the selective activation of this pathway in the cell under stress. Similarly to what we observed for abiotic stress, some EXO70 isoforms were found to be upregulated under biotic stress conditions [57]. As to what has been described for VTI12, the role of this protein in the autophagic process and/or in EXPO could be important for the protein degradation and membrane remodelling occurring during stress events. To better understand the impact of stress in these autophagic processes, it would be interesting to check the expression pattern of other genes, such as different exocyst subunits, other members of the ATG8 family and other V-snares belonging to the VTI12 family.

### 3.3. An Initial Draft of Endomembrane System Response to Stress

It is known that adverse environmental conditions challenge the cell’s endomembrane organisation that needs to readjust protein biogenesis and trafficking pathways. In the present study, we analysed the cell ultrastructure and the expression of several genes involved in the vacuolar pathway and exocytosis/autophagy routes during stress conditions. These results are summarized in Figure 7, which depicts a cell in the control condition (Figure 7—left panel) and a cell under stress conditions (Figure 7—right panel). the data obtained allow us to conclude that the endomembrane system suffers alterations in the morphology of some organelles, such as the endoplasmic reticulum and Golgi apparatus, and a proliferation of vesicles can be observed in some conditions confirming the high activity of trafficking mechanisms in these cells (Figure 7—right panel, orange-labelled organelles). Regarding the analysis of the expression of the selected genes, we can conclude that the vacuolar pathway to the protein storage vacuole is favoured, contrary to the route to the lytic vacuole, as evidenced by the expression of vacuolar receptors and members of the SNARE family involved (Figure 7—right panel, green and red arrows). Additionally, the pathway mediated by the exocyst seems to be enhanced as well, while the autophagic route does not have major alterations, confirming the important role of plant defence systems in response to abiotic stress. To further explore the data presented here, it would be interesting to test the expression and localization of other genes involved in the described pathways and perform genetic tests with select genes.

Responses of plants to stress are not linear but are a very complex network of integrated circuits involving multiple pathways, cellular compartments, and the interaction of signalling molecules to give coordinated responses to different stimuli. The underlying mechanisms are far from being uncovered. The whole picture needs to integrate the environmental signals that lead to an appropriate response in the adjustments of endomembrane trafficking and take into consideration the response to the crosstalk of multiple stresses. Facing the adverse effects of various abiotic stresses on agricultural production is imperative to identify and characterize critical genes involved in plant stress responses. Wang et al. [11] go further, proposing to engineer stress-tolerant crops, namely by manipulating certain regulatory genes controlling the expression of many stress-responsive genes. Stress-inducible promoters could be used in genetically engineered plants to improve tolerance to abiotic stresses in crops. These authors argue that in the future stress biotechnology research should rely on the stress-induced expression of the transgenes, the regulatory machinery, and the use of transcription factors for controlling the expression of stress-responsive genes. As such, this work will serve as a basis for more detailed studies and contribute to drawing a mechanistic view of the alterations and responses of the endomembrane system during abiotic stress.

## 4. Materials and Methods

### 4.1. Biological Material

*Arabidopsis thaliana* (col0) seeds were germinated in Murashige and Skoog medium (MS) (Duchefa), supplemented with 1.5% (*w*/*v*) sucrose and 0.7% (*w*/*v*) bactoagar. For the abiotic stress conditions, different concentrations of stress inductors were added to the medium as indicated in Table 2: sodium chloride (saline stress), mannitol (hydric stress), hydrogen peroxide (oxidative stress) and zinc sulphate (heavy metal-induced stress). Seeds were germinated and maintained in a growth chamber for ten to twelve days, at 21 °C with a photoperiod of 16 h light and average humidity between 50 and 60%. Three biological replicates were prepared.

### 4.2. Electron Microscopy

Small pieces of leaves from seedlings exposed to different stress conditions (as described before) were cut into pieces of approximately 1–2 mm for transmission electron microscopy. These were fixed in NaPIPES (1.25% (*w*/*v*) and pH 7.2) with 2.5% (*v*/*v*) glutardialdehyde for one hour at room temperature (RT). Then, the samples were washed with 2.5% (*w*/*v*) NaPIPES (3 times for 10 min each) and post-fixed in 4% (*w*/*v*) osmium tetroxide prepared in 2.5% (*w*/*v*) NaPIPES for 2 h at RT. The samples were dehydrated in a graded series of increasing concentrations of ethanol with incubation times of 10 min (10%), 15 min (20, 30 and 50%) and 20 min (70 and 100%). The ethanol was then exchanged with propylene oxide for 20 min. The specimens were gradually infiltrated in increasing concentrations of Epoxy resin (Agar scientific): 10% (for 15 min) to 50% (overnight) and 100% (overnight). The tissue pieces were then transferred to oriented moulds, covered with resin, and polymerized at 60 °C overnight. Ultrathin sections (60–80 nm) were cut in UC6 Ultramicrotome (Leica, Carnaxide, Portugal), recovered to 400 mesh copper grids, post-stained with Uranyless EM Strain (Uranyless) and with Reynolds Lead citrate 3% (Uranyless) for 5 min each. After washing to remove excess stain, grids were observed in a Jeol JEM 1400 Transmission Electron Microscope, and the images were acquired with an Orius DC200D camera (Gatan, Pleasanton, CA, USA). Image processing was done using ImageJ/Fiji software.

### 4.3. cDNA Preparation

Total RNA was prepared using the “NZY Total RNA Isolation Kit” (NZYTech, Lisboa, Portugal) according to the manufacturer instructions and starting from 100 mg of seedling tissue. Three biological replicates were prepared, and samples were quantified using a Nanodrop spectrophotometer (DeNovix DS-11, Bonsai Lab, Madrid, Spain). RNA integrity was verified in a 1% (*w*/*v*) agarose gel. A SuperScript IV VILO Master Mix (Thermo Fisher Scientific, Waltham, MA, USA) kit was used to obtain cDNA preparations, following the protocol provided. Both RNA and cDNA were kept at −80 °C until use.

### 4.4. Gene Selection and Interaction Map

The selection of the genes for this study was based on their product localization, role, interaction partners and literature references (Table 1). The interaction mapping of the different gene products was obtained using the online software String (https://version-11-0b.string-db.org/, accessed on 20 July 2021), based on protein–protein interaction networks functional enrichment analysis, where the line thickness indicates the strength of data support [35].

### 4.5. Quantitative RT-PCR

For the quantitative RT-PCR, three biological replicates and three technical replicates were performed for each gene and condition. The reaction was performed in a CFX96 Real-Time System (BioRad, Hercules, CA, USA) using PowerUp SYBR Green Master Mix (Thermo Fisher Scientific). In brief, the 10 µL reaction included 400 nM of each primer (Table 3) and 2 µL of cDNA in a 1:8 dilution. The amplification conditions included an initial denaturation (95 °C for 3 min), followed by 40 cycles of amplification and quantification (95 °C for 10 s, 56 °C for 10 s and 72 °C for 30 s with a single fluorescence measurement) and melting curve generation (65 °C to 95 °C with one fluorescence read every 0.5 °C). Calculation of cycle threshold (Ct), primer efficiency and expression tests were performed using Bio-Rad CFX Maestro (version 1.0) software. The analysis of the results obtained was carried out by comparison with the control condition. The housekeeping genes SAND-1 and UBC9 were used as reference genes, based on the study of Czechowski and collaborators [58]. The gene clustering by average linkage and heat map was obtained using the online software Heatmapper (http://www.heatmapper.ca/, accessed on 3 August 2021), applying the Pearson correlation between the normalized expression of the different genes and the stress conditions [59].

## Figures and Tables

**Figure 1 ijms-22-10644-f001:**
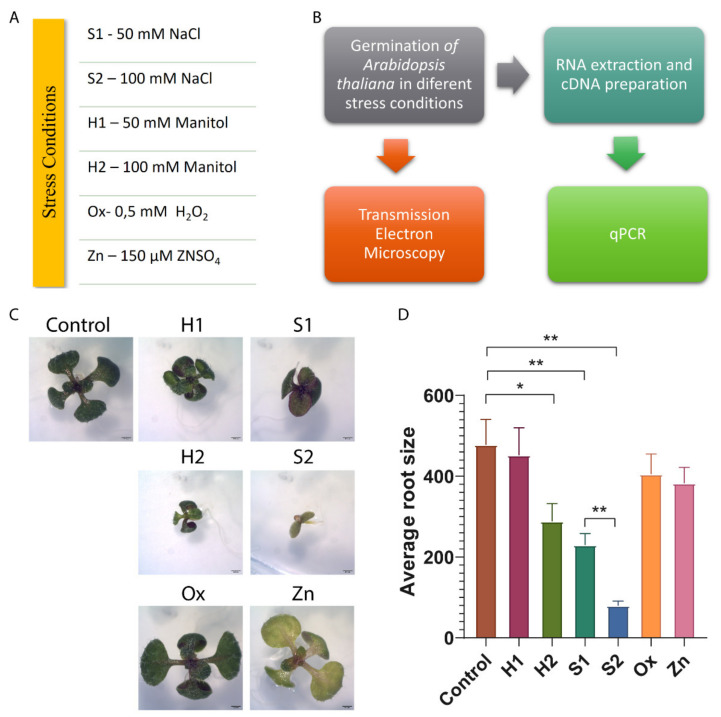
*Arabidopsis thaliana* seedling growth conditions, workflow schematic and biometric analysis. (**A**) Medium conditions used to induce stress: S1—salt stress, low concentration; S2—salt stress, high concentration; H1—hydric stress, low concentration; H2—hydric stress, high concentration; Ox—oxidative stress; Zn—metal stress induced by zinc. (**B**) Schematic representation of the analysis performed. (**C**) *Arabidopsis thaliana* seedlings size comparison in stress conditions. (**D**) Bar plot of the average measured root size in stress conditions relative to control. * Indicates statistically experimental values (*p* < 0.05) while ** indicates statistically experimental values (*p* < 0.1). Scale bars: (**C**) 0.1 cm.

**Figure 2 ijms-22-10644-f002:**
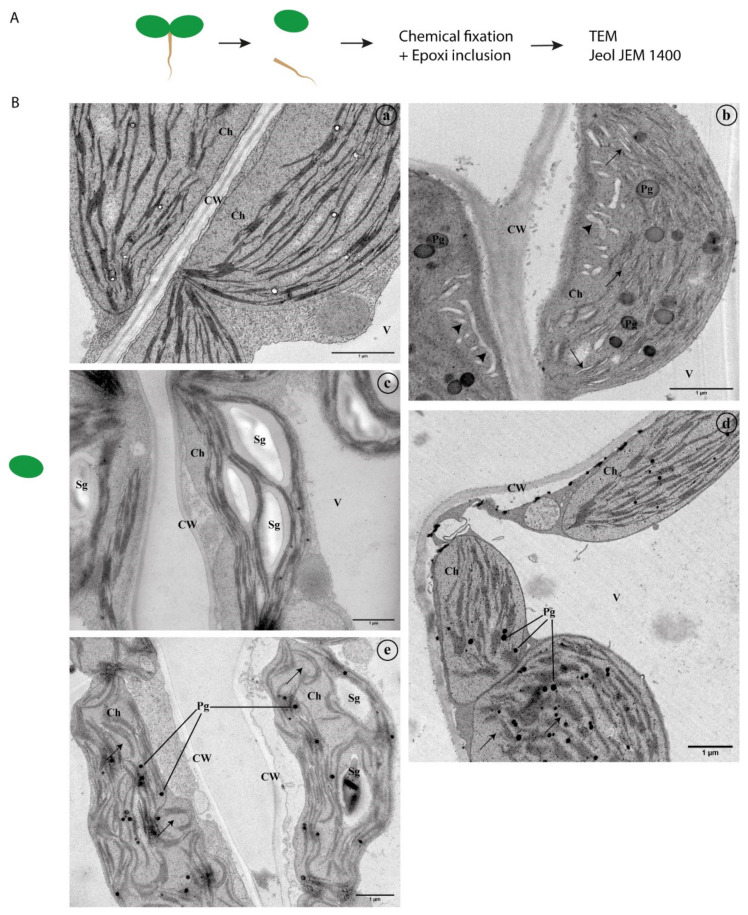
Transmission electron microscopy analysis of *A. thaliana* leaf cells. (**A**) Workflow to perform transmission electron microscopy. (**B**) *A. thaliana* leaf cells visualisation under different stress conditions. (**a**)—Control; (**b**)—Saline Stress (100 mM NaCl); (**c**)—Hydric stress (100 mM NaCl); (**d**)—Oxidative stress; (**e**)—Zinc stress. Abbreviations: CW—Cell wall; Ch—Chloroplast; Pg—Plastoglobuli; Sg—Starch granules; V—Vacuole. Scale bars: 1 μm.

**Figure 3 ijms-22-10644-f003:**
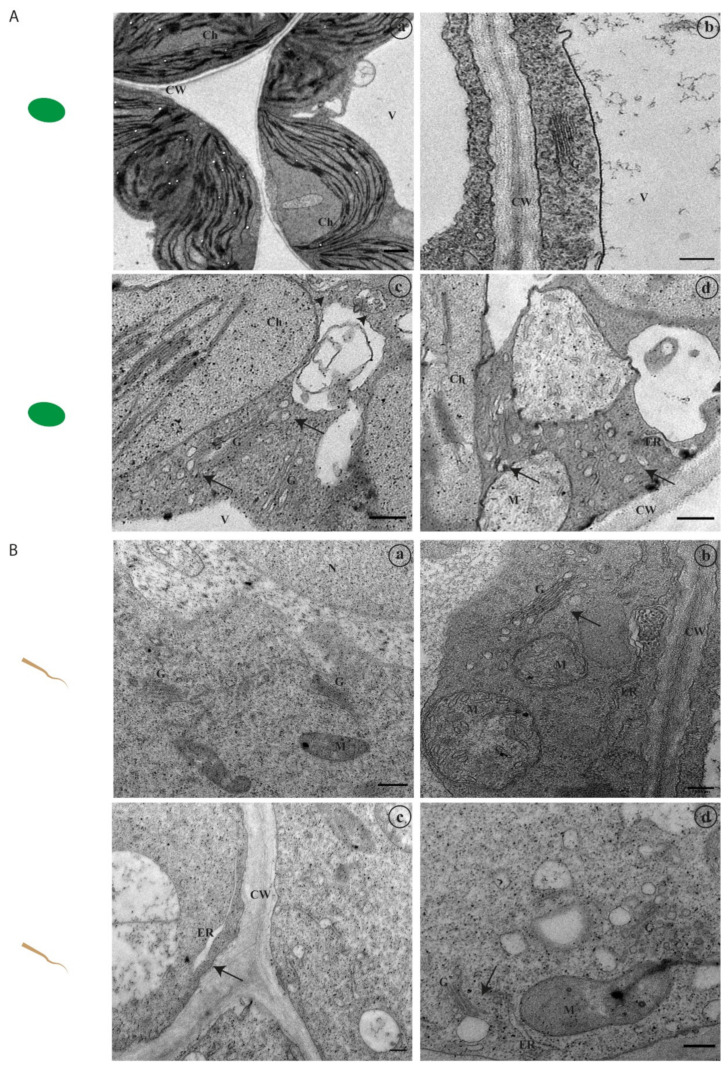
Transmission electron microscopy analysis of *A. thaliana* leaf and root cells. (**A**) *A. thaliana* leaf cells visualisation and comparison between control condition (**a**,**b**) and oxidative stress condition (**c**,**d**). (**B**) *A. thaliana* root cells visualisation and comparison between control condition (**a**) and hydric stress (**b**) and oxidative stress condition (**c**,**d**). Abbreviations: CW—Cell wall; Ch—Chloroplast; ER—Endoplasmic Reticulum; G—Golgi, M—Mitochondria. Scale bars: (**A**) (**a**)—0.8 μm. (**b**)—200 nm. (**c**,**d**)—0.2 μm; (**B**) (**a**,**c**,**d**)—0.3 μm. (**b**)—150 nm.

**Figure 4 ijms-22-10644-f004:**
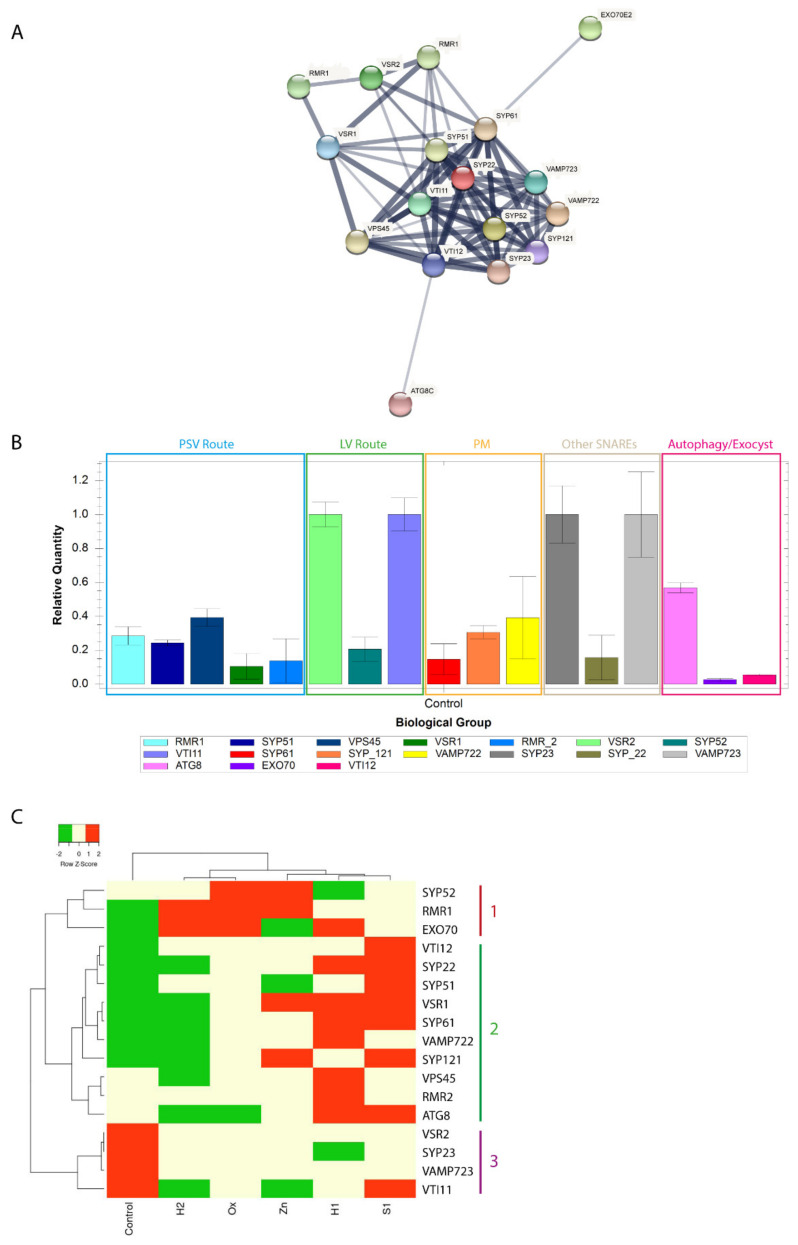
Differential analysis of the genes selected for the study. (**A**) Interaction network of the gene products selected in the study obtained using the String online software (https://version-11-0b.string-db.org/, accessed on 20 July 2021). Line thickness indicates the strength of data support; (**B**) heatmap of the Pearson correlation of the normalized gene expression data clustered by gene (rows) and stress (columns), obtained using the online software Heatmapper (http://www.heatmapper.ca/, accessed on 3 August 2021); (**C**) bar chart representing the relative quantity of all the genes under study in control condition. Coloured boxes represent the different groups organized for qRT-PCR analysis.

**Figure 5 ijms-22-10644-f005:**
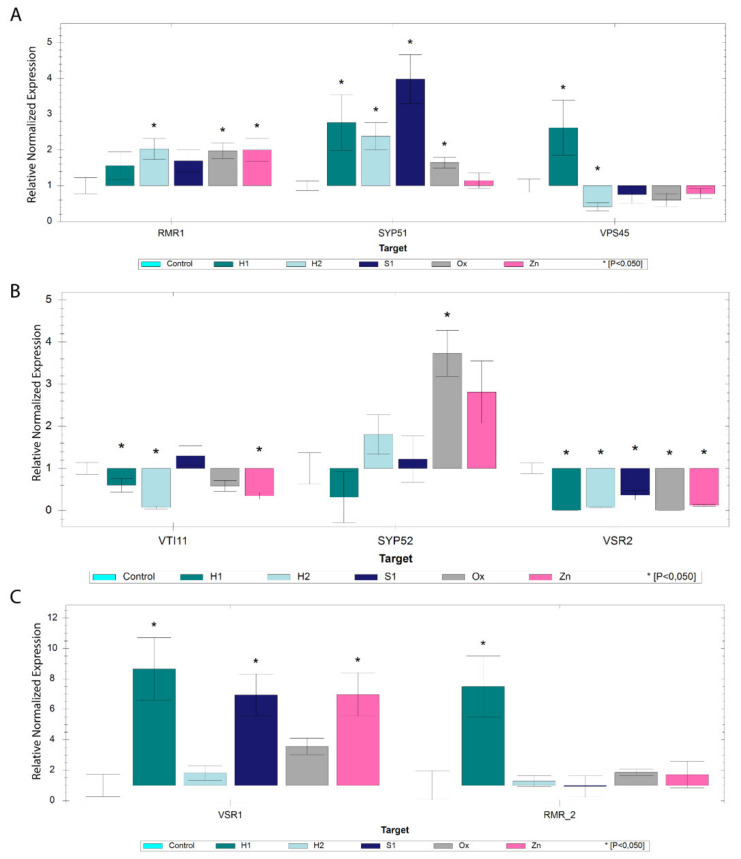
Expression analysis by qRT-PCR of genes involved in vacuolar routes. (**A**) Bar chart analysis of *AtRMR1*, *AtSYP51* and *AtVPS45* genes involved in trafficking to the protein storage vacuole in different stress conditions relative to control. (**B**) Bar chart analysis of *AtVSR2*, *AtVTI11* and *AtSYP52* genes involved in trafficking to the protein storage vacuole in different stress conditions relative to control. (**C**) Bar chart analysis of *AtRMR2* and *AtVSR1* genes involved in trafficking to the protein storage vacuole in different stress conditions relative to control. * indicates statistically experimental values (*p* < 0.05).

**Figure 6 ijms-22-10644-f006:**
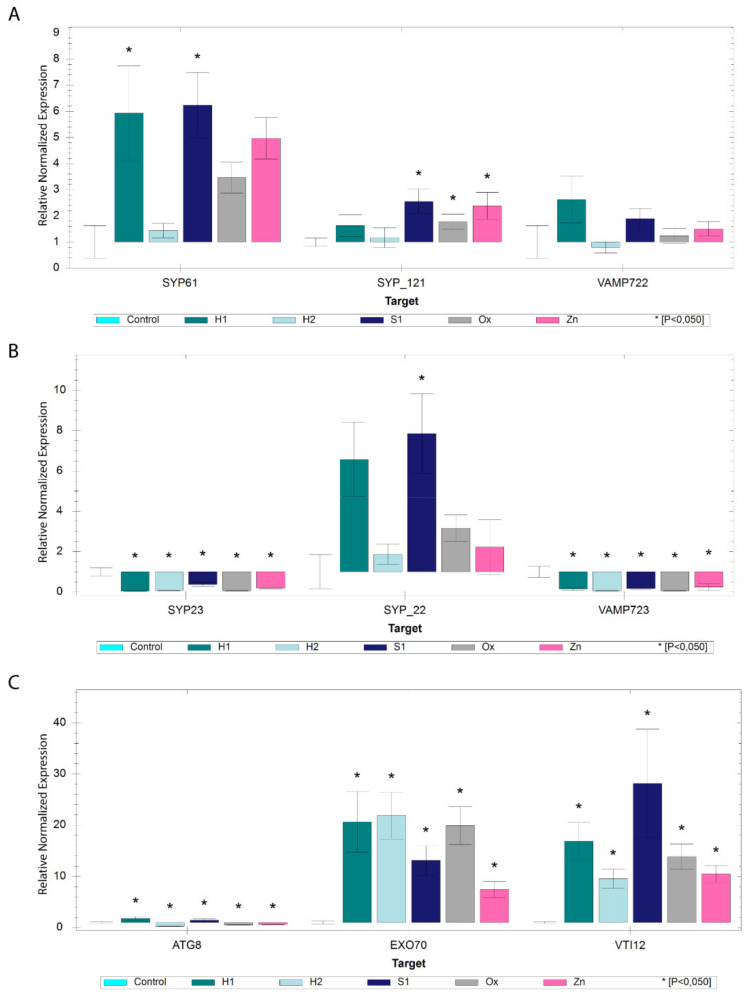
Expression analysis by qRT-PCR of genes involved in plasma trafficking and autophagy/exocyst-mediated routes. (**A**) Bar chart analysis of *AtSYP61*, *AtSYP121* and *AtVAMP722* genes involved in cell membrane trafficking processes in different stress conditions relative to control. (**B**) Bar chart analysis of *AtVAMP723*, *AtSYP23* and *AtSYP22* genes in different stress conditions relative to control. (**C**) Bar chart analysis of *AtExo70*, *AtVTI12* and *AtATG8* genes involved in exocytosis and autophagy processes in different stress conditions relative to control. * indicates statistically experimental values (*p* < 0.05).

**Figure 7 ijms-22-10644-f007:**
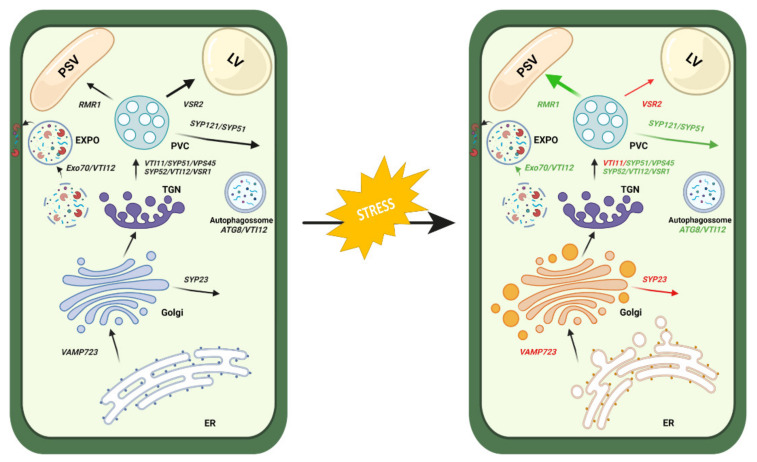
Schematic representation of the plant cell in control versus stress conditions. **Left panel**—control cell; **right panel**—cell under stress. The localization of the molecular players analysed in the context of this study is indicated and shown in red or green according to their expression profile (down- or upregulated, respectively). Black arrows represent the different pathways analysed, red arrows indicate downregulated routes and green arrows correspond to upregulated routes. In the right panel, organelles labelled in orange represent the ultrastructural changes observed in relation to control. Image created with BioRender.com, accessed on 28 September 2021.

**Table 1 ijms-22-10644-t001:** Endomembrane-related genes used in this study.

Gene	Identifier	Role and Localisation	References
*AtVSR2*	AT1G71980.1	Involved in the trafficking of vacuolar proteins. May function as a sorting receptor for protein trafficking to the protein storage vacuole (PSV).	[20]
*AtATG8*	AT1G62040.2	Ubiquitin-like modifier involved in autophagosome formation. May mediate the delivery of the autophagosomes to the vacuole via the microtubule cytoskeleton; belongs to the ATG8 family	[21]
*AtEXO70*	AT5G61010.1	Essential component for the formation and the recruitment of exocyst subunits to the exocyst-positive organelle (EXPO), a secreted double membrane structure also called extracellular exosome.	[22]
*AtVAMP722*	AT2G33120.2	Involved in the targeting and/or fusion of transport vesicles to their target membrane. Associated with biotic stress response. Defines a complex with SYP121.	[23,24]
*AtRMR1*	AT5G66160.1	Involved in the trafficking of vacuolar proteins. Functions probably as a sorting receptor for protein trafficking to the protein storage vacuole (PSV) by binding the C-terminal vacuolar sorting determinant (VSD) of vacuolar-sorted proteins	[25]
*AtVTI11*	AT5G39510.1	May function as a v-SNARE responsible for targeting AtELP-containing vesicles from the trans-Golgi network (TGN) to the prevacuolar compartment (PVC). Promotes the formation of vacuolar membrane ‘bulbs’.	[26]
*AtSYP121*	AT3G11820.1	Encodes a syntaxin localized at the plasma membrane. Functions in positioning anchoring of the KAT1 K+ channel protein at the plasma membrane. Additionally, functions in non-host resistance against barley powdery mildew. Forms a complex with VTI11 and SYP51.	[10,27]
*AtSYP23*	AT4G17730.2	Cytosolic syntaxin with no transmembrane domain. May function in the docking or fusion of transport vesicles with the prevacuolar membrane.	[28]
*AtSYP51*	AT1G16240.1	Involved in the transport of chitinase to the PSV. Interacts with VTI12.	[29]
*AtSYP52*	AT1G79590.2	Involved in the route to the Lytic vacuole. Forms a complex with VTI11.	[29]
*AtSYP61*	AT1G28490.1	Vesicle trafficking protein that functions in the secretory pathway. Involved in osmotic stress tolerance and in abscisic acid (ABA) regulation of stomatal responses. Forms a complex with VTI12.	[10]
*AtSYP22*	AT5G46860.1	Syntaxin-related protein required for vacuolar assembly. Localized in the vacuolar membranes.	[28]
*AtVAMP723*	AT2G33110.1	Localized at the endoplasmic reticulum. Involved in the targeting and/or fusion of transport vesicles to their target membrane.	[30]
*AtVPS45*	AT1G77140.1	A peripheral membrane protein that associates with microsomal membranes, likely to function in the transport of proteins to the vacuole. Has been described as a positive regulator for PSV route. Forms a complex with VTI12 and SYP61.	[31]
*AtVSR1*	AT3G52850.1	Vacuolar-sorting receptor (VSR) involved in clathrin-coated vesicles sorting from Golgi apparatus to vacuoles. Required for the sorting of 12S globulin, 2S albumin and maybe other seed storage proteins to protein storage vacuoles (PSVs) in seeds.	[32]
*AtVTI12*	AT1G26670.1	Normally localizes to the trans-golgi network and plasma membrane. Involved in protein trafficking to protein storage vacuoles	[26,33]
*AtVSR2*	AT2G14720.1	Vacuolar sorting receptor (VSR) involved in clathrin-coated vesicles sorting from Golgi apparatus to vacuoles; belongs to the VSR (BP-80) family.	[34]

**Table 2 ijms-22-10644-t002:** Different conditions and additive concentrations used to induce stress during germinating *Arabidopsis thaliana* seeds.

Stress Type	Nomenclature	Additive	Concentration
Control	Control	-	-
Saline	S1	Sodium Chloride	50 mM
	S2	Sodium Chloride	100 mM
Hydric	H1	Mannitol	50 mM
	H2	Mannitol	100 mM
Oxidative	Ox	Hydrogen Peroxide	0.5 mM
Heavy Metal	Zn	Zinc Sulphate	150 µM

**Table 3 ijms-22-10644-t003:** List of genes and corresponding primer pairs used in the quantitative RT-PCR assay.

Genes	Primer Forward	Primer Reverse
*AtSYP23*	GCAGCGTGCCCTTCTTGTGG	TCCTTGGGCAGTTGCAGCGTA
*AtSYP121*	TCCTCCGATCGAACCAGGACCTC	TTCTCGCCGGTGACGGTGAA
*AtVAMP723*	CCCGTGGTGTGATATGTGAG	CCACAAACCGAGAGGATGAT
*AtVTI12*	GCAATGTCCGTGGAGAGGCTTGA	TGCGCATGAAGGAGGGTTTGG
*AtBP-80*	GGGAGCGGCGCAGATTCTTG	GCCGGTTTCATTCGCCACCTT
*AtRMR1*	GCGAGGGAGGCACACCAGGA	TTTCCCCGGCCTTGTGGTGA
*AtEXO70*	TCCCCGATGAAACAGGCTCGTC	GCCTCCATGAAAGGGGCGTGT
*AtUBC9* ^1^	TCACAATTTCCAAGGTGCTGC	TCATCTGGGTTTGGATCCGT
*AtSAND-1* ^1^	AACTCTATGCAGCATTTGATCCACT	TGATTGCATATCTTTATCGCCATC
*AtVTI11*	GATTTTCCGACTCCGACGTA	CAAACGCGTCACTCATTGTT
*AtSYP52*	ATGTGGTGGCAACTTGTGAA	CTTTGCCTCACAGACACGAA
*AtVPS45*	CAGTTGTCGATCCCTCTGGT	TTTCAACGCCAGAAACACTG
*AtSYP61*	TTGAAAAACGGAGGAGATGG	TTCACTTGCATGACCTGCTC
*AtVAMP722*	CAATTTGTGGGGGATTCAAC	GATCTTGGGAAGCACAGAGC
*AtVSR1*	GATGTGGACGAGTGCAAAGA	GCTCACGCATGTAAAGCAAA
*AtRMR2*	CTTCCTTGGGCTCATTACCA	GGTGATCGACGGTTAGAGGA

^1^ from (Czechowsaki et al., 2005).

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
