# Peer review of "Abiotic Stress Triggers the Expression of Genes Involved in Protein Storage Vacuole and Exocyst-Mediated Routes"

_ijms, 2021, doi:10.3390/ijms221910644_

Round 1

Reviewer 1 Report

The manuscript entitled “Abiotic Stress triggers the Expression of Genes Involved in PSV and Exocyst-mediated Routes” is interesting one and here are some comments on this.

Please revise the manuscript for spelling and grammatical errors. The presentation of the results and discussion section could be simpler for better understanding.  

Line 28: “Climate change stands, nowadays, as the foremost threat to human health” How?

Line 32-33: “Drought, salinity, temperature and oxidative stress are often interrelated and 32 cause similar cellular damage [3].” Oxidative stress is the secondary stress caused by different stresses. Please rewrite considering this.

At the end of Introduction the objectives should be described clearly.

The Results section is too elaborated and sometimes difficult to understand. I suggest presenting in simpler way and in different paragraph (where applicable) for better understanding.

In Fig 1: The caption/title should explain all the treatments including H1, H2, S1, S2.

Check the spelling Fid 1D, Y axis: size or size. “average root syze”. Please rewrite considering capitalization and spelling.

Fig. 1A Please check “0,5 mM H2O2

Fig 1C: Phenotypically the control and Ox treated plants look similar, what is the reason?

Fig 1, 2 and other figures and all the cases: caption/title: A. thaliana should be Italic

2.2.1. Ultrastructural characterization in leaf tissues: here different location and organelles can be described in different paragraph for clear understanding

Line 320: “The major changes in the endomembrane system were observed in the oxidative  stress condition.” But the Fig. 1 phenotypic appearance doesn’t indicate that the oxidative stress created the worse condition.

Line 362: “genes involved in autophagy/exocytosis routes” how these are related to the studied stress conditions of the present study please explain.

Fig. 7: This figure should be self explanatory. What is indicated by this comparative picture it should be described briefly in the figure caption. What does it mean by different colors in the figure it should also be mentioned.

Line 506-511: It is good that the authors presented a comparative picture in Fig 7. But the explanation should be more clarified. The mechanism can be explained in better way for clear understanding.

At the end of discussion it is suggested to explain how different Abiotic Stresses (saline, hydric, oxidative, heavy metal) of the present study triggered the Expression of Genes Involved in 2 PSV and Exocyst-mediated Routes and how all these are involved and can be implicated for improving the plant abiotic stress tolerance should be described in brief.  

Table 2: Please check “0.,5 mM” of last column.

Line 541: Please check the spelling “over-nigh.”

Please include the references of these mythologies: 4.2. Electron Microscopy; 4.3. cDNA preparation

Author Response

Thank you very much for the critical reading of our manuscript and all the comments and suggestions made, which will certainly improve the quality of our manuscript. All comments were taken into consideration and the text was corrected accordingly and/or a rationale explaining our point of view is provided. The entire manuscript was revised for the correction of spelling and grammatical errors. All the comments and suggestions were addressed point-by-point as follows:

  • Please revise the manuscript for spelling and grammatical errors.
    • Reply: The entire manuscript has been corrected for grammatical errors and spelling mistakes.
  • The presentation of the results and discussion section could be simpler for better understanding. 
    • Reply: The English revision of the document allowed to simplify some sentences, which make the text more comprehensible, and all the corrections suggested by the reviewer in this section were made.
  • Line 28: “Climate change stands, nowadays, as the foremost threat to human health” How?
    • Reply: The text was changed to clarify this point.
  • Line 32-33: “Drought, salinity, temperature, and oxidative stress are often interrelated and 32 cause similar cellular damage [3].” Oxidative stress is the secondary stress caused by different stresses. Please rewrite considering this.
    • Reply: The text was changed to clarify this point.
  • .At the end of the Introduction the objectives should be described clearly.
    • Reply: The end of the Introduction section was changed, and the objectives are now clearly presented.
  • The Results section is too elaborated and sometimes difficult to understand. I suggest presenting in a simpler way and in different paragraphs (where applicable) for better understanding.
    • Reply: Several paragraphs were added in the results section to separate the different subjects.
  • In Fig 1: The caption/title should explain all the treatments including H1, H2, S1, S2.
    • Reply: The caption of the figure was corrected.
  • Check the spelling Fid 1D, Y-axis: size or size. “average root syze”. Please rewrite considering capitalization and spelling.
    • Reply: The spelling in figure 1D was corrected.
  • 1A Please check “0,5 mM H2O2
    • Reply: This point was corrected in Figure 1A.
  • Fig 1C: Phenotypically the control and Ox treated plants look similar, what is the reason?
    • Reply: It is true that in terms of development the control and Ox treatment look similar. One possible explanation is that the concentration of the stress inductor used in this case is not sufficient to impair the normal development of the plants. Moreover, the alterations observed at the ultrastructural level possibly do not interfere with the development of the plant and may indicate, instead, a response of the cell to cope with the adverse condition it is facing. These hypotheses were included in the discussion section.
  • Fig 1, 2 and other figures and all the cases: caption/title:  thalianashould be Italic
    • Reply: All figure captions were corrected.
  • 2.1. Ultrastructural characterization in leaf tissues: here different locations and organelles can be described in different paragraphs for a clear understanding
    • Reply: As mentioned above, several paragraphs were included in the results section for simplification.
  • Line 320: “The major changes in the endomembrane system were observed in the oxidative  stress condition.” But the Fig. 1 phenotypic appearance doesn’t indicate that the oxidative stress created the worse condition.
    • Reply: This has already been addressed above and a possible justification is provided in the discussion section.
  • Line 362: “genes involved in autophagy/exocytosis routes” how these are related to the studied stress conditions of the present study please explain.
    • Reply: The relation between these two genes and stress is explained in the section “Exocytosis and/or autophagy mediated by the exocyst is stimulated by stress”, which was improved with more details regarding the stress conditions used.
  • 7: This figure should be self-explanatory. What is indicated by this comparative picture it should be described briefly in the figure caption. What does it mean by different colors in the figure it should also be mentioned.
    • Reply: The image caption was changed to include more details.
  • Line 506-511: It is good that the authors presented a comparative picture in Fig 7. But the explanation should be more clarified. The mechanism can be explained in better way for clear understanding.
    • Reply: More details were added in this section to better explain the Figure.
  • At the end of discussion it is suggested to explain how different Abiotic Stresses (saline, hydric, oxidative, heavy metal) of the present study triggered the Expression of Genes Involved in 2 PSV and Exocyst-mediated Routes and how all these are involved and can be implicated for improving the plant abiotic stress tolerance should be described in brief.  
    • Reply: A new paragraph was included at the end of the discussion section to address these points.
  • Table 2: Please check “0.,5 mM” of last column.
    • Reply: Table 2 was corrected.
  • Line 541: Please check the spelling “over-nigh.”
    • Reply: The spelling was corrected.
  • Please include the references of these mythologies: 4.2. Electron Microscopy; 4.3. cDNA preparation
    • Reply: The protocols for these two techniques are described in detail and are routinely used in the laboratory, thus there is no need to add a reference.  

Reviewer 2 Report

Comments:

This manuscript is well written to report a uthors’ research testing the hypothesis that plants respond to stress by remodeling their endomembranes and adapting their trafficking pathways. They used transmission electron microscopy to observe the ultrastructural cell alterations in leaf and root tissues. They also investigated the expression of several endomembrane system-associated genes involved in the vacuolar pathway and exocytosis/autophagy routes during stress conditions. They concluded that the vacuolar pathway to the protein storage vacuole is favored, contrary to the route to the lytic vacuole. The schematic representation of the plant cell in control versus stress conditions in Figure 7 is interesting.

Suggestions:

  1. For all Tables, make each cell (field) top- and left-justified to allow easy reading and understanding.
  2. Plural form should be used for (Figures 2 and 3), (Figures 2A and 3A), etc.
  3. Wherever appropriate, “stress” should be changed to “stresses”.

Author Response

Thank you very much for the critical reading of our manuscript and all the comments and suggestions made, which will certainly improve the quality of our manuscript. All comments were taken into consideration and the text was corrected accordingly. The entire manuscript was revised for the correction of spelling and grammatical errors. All the comments and suggestions were addressed point-by-point as follows:

Suggestions:

  • For all Tables, make each cell (field) top- and left-justified to allow easy reading and understanding.
    • Reply: All tables were adjusted accordingly to what was suggested.
  • Plural form should be used for (Figures 2 and 3), (Figures 2A and 3A), etc.
    • Reply: The manuscript text was altered, and the plural form was used where appropriate.
  • Wherever appropriate, “stress” should be changed to “stresses”.
    • Reply: This change was made along the entire document whenever was appropriate.

Reviewer 3 Report

The authors of the current manuscript studied the impacts of several abiotic stress conditions on the ultrastructure and operation of the endomembrane system in Arabidopsis cells. Their major findings show that stress alters cellular ultrastructure in leaves and roots and leads to differential expression of a set of genes involved in protein trafficking.

The study is interesting and novel. However, there are some considerations that may need to be addressed in revisions to improve the manuscript:

  1. Clarify whether the used stress conditions represent extreme strong or mild stress, whether they are conditions potentially experienced by plants under field conditions.
  2. The title may need reformulating (spell out what PSV stands for). As is, the title seems unfinished. …”PSV and exocyst-mediated routes for…..”.
  3. Lines 20-22, when referencing the activation and deactivation of “certain pathways” – please specify.
  4. Lines 336-338 when stating: “The high amount of storage of substances found in these samples, in particular carbohydrates, can be seen as a result of a greater intracellular compartmentalization [42] in order to be able to manage the water deficit to which the cells are subject” – I recommend revising this statement. The accumulation of elevated amounts of starch under the drought conditions used in this study may be due to the impairment of sugar metabolism (may indicate a decrease in cellular respiration…).
  5. Figure 7 needs a more detailed figure description. Please note what the differently colored text, arrows, arrow width, etc. represent (upregulation, downregulation, etc. ?).
  6. Figure 1D change “syze” to "size" on the Y axis label.
  7. In the text the authors frequently use the term “situation” to reference the control or stress conditions used for the study. I recommend revising the text and replacing “situation” with “condition”.
  8. Overall, I think the authors should make a better connection with what is known about plant responses to the used stress conditions, at the strength they applied the stress at, in the Discussion. The mechanisms of these stresses have all been studied extensively, and would benefit the paper if the novel data collected in this study would better integrate into the context of plant stress responses (stress induced damages and stress acclimation and adaptation).

Author Response

Thank you very much for the critical reading of our manuscript and all the comments and suggestions made, which will certainly improve the quality of our manuscript. All comments were taken into consideration and the text was corrected accordingly and/or a rationale explaining our point of view is provided. The entire manuscript was revised for the correction of spelling and grammatical errors. All the comments and suggestions were addressed point-by-point as follows:

  • Clarify whether the used stress conditions represent extreme strong or mild stress, whether they are conditions potentially experienced by plants under field conditions.
    • Reply: The stress conditions used were selected from the literature available on Arabidopsis plants under stress and the ones used correspond to high concentrations of the stress agent, to have a visible effect, but not severe that would impair seed germination and seedling viability.

  • The title may need reformulating (spell out what PSV stands for). As is, the title seems unfinished. …”PSV and exocyst-mediated routes for…..”.
    • Reply: The authors do not agree with this comment, because the title is quite clear. Nevertheless, as the reviewer suggested, the PSV abbreviation was removed and is now full-length.

  • Lines 20-22, when referencing the activation and deactivation of “certain pathways” – please specify.
    • Reply: In this sentence, the authors mean the intracellular sorting pathways, including vacuolar, secretory and autophagic routes. This information was included in the text.

  • Lines 336-338 when stating: “The high amount of storage of substances found in these samples, in particular carbohydrates, can be seen as a result of a greater intracellular compartmentalization [42] in order to be able to manage the water deficit to which the cells are subject” – I recommend revising this statement. The accumulation of elevated amounts of starch under the drought conditions used in this study may be due to the impairment of sugar metabolism (may indicate a decrease in cellular respiration…).
    • Reply: The authors thank the reviewer for the clarification regarding this issue and the information was added in the text.

  • Figure 7 needs a more detailed figure description. Please note what the differently colored text, arrows, arrow width, etc. represent (upregulation, downregulation, etc.?).
    • Reply: The caption of figure 7 was changed and more details concerning the figure were included.
  • Figure 1D change “syze” to "size" on the Y axis label.
    • Reply: Figure 1D was corrected.
  • In the text the authors frequently use the term “situation” to reference the control or stress conditions used for the study. I recommend revising the text and replacing “situation” with “condition”.
    • Reply: The text was revised and corrected according to the reviewer's suggestion.

  • Overall, I think the authors should make a better connection with what is known about plant responses to the used stress conditions, at the strength they applied the stress at, in the Discussion. The mechanisms of these stresses have all been studied extensively and would benefit the paper if the novel data collected in this study would better integrate into the context of plant stress responses (stress induced damages and stress acclimation and adaptation).
    • Reply: Plant responses to stress (both stress-induced damage and plant adaptation), have been thoroughly studied over the past decades and several good reviews on the subject are available. This is addressed in the introduction section and the discussion section whenever felt necessary to justify the results obtained. Moreover, the effects of the stress in the physiology of the plants and the biochemical mechanisms underneath are not the focus of this work and we consider that the data presented pave the way for further studies.